# Simulation and Experimental Verification of the Thermal Behaviour of Self-Written Waveguides

Axel Günther [1,2,*] , Murat Baran [3] , Wolfgang Kowalsky [1,2] and Bernhard Roth [2,3]

1 Institute of High Frequency Technology, Technical University Braunschweig, Schleinitzstraße 22, 38106 Braunschweig, Germany; wolfgang.kowalsky@ihf.tu-bs.de
2 Cluster of Excellence PhoenixD (Photonics, Optics and Engineering—Innovation Across Disciplines), 30167 Hannover, Germany; Bernhard.Roth@hot.uni-hannover.de
3 Hannover Centre for Optical Technologies, Leibniz University of Hannover, Nienburger Str. 17, 30167 Hannover, Germany; murat.baran@hot.uni-hannover.de
* Correspondence: axel.guenther@ihf.tu-bs.de

**Abstract:** In this work, we investigated the optical response of a self-written waveguide (SWW) in detail by heating the structure from room temperature up to 60 °C. Previous results indicated a decrease in the optical transmission with increasing temperature for certain waveguide parameters. Based on new experimental measurements, we have identified material parameters resulting in opposite behaviour. An experimental setup was conceived to verify these results. Hereby, we were able to show that we can adjust material parameters such as refractive index and the corresponding density of the material by adapting the curing time applied during the fabrication of the waveguides. This, in turn, affects the material's response during the heating process. We showed that a limitation of the external curing time changes the internal conditions of the SWW and the cladding in a manner that the numerical aperture increases with the temperature, which subsequently also results in an increase in the optical transmission. In this study, we explain this unexpected behavior of the SWW and point towards possible future applications.

**Keywords:** optical interconnects; self-written waveguides; optical simulation; refractive index measurements; thermal simulation; optical sensor





## 1. Introduction

Photonic structures and components have become extremely important during the last decades. Due to higher data transmission and electromagnetic compatibility, they can supersede their electronic counterparts for special applications, i.e., in a biomedical context or in high voltage environment [1,2]. In particular, polymer-based optical components and systems have become more attractive due to the lower production costs compared to their semiconductors or glass-based counterparts [3]. Next to pricing, polymers offer multiple characteristics which make them suitable for, e.g., optical sensing and short-distance transmission. Additional important aspects for photonic structures are optical interconnects to overcome gaps between optical waveguides or for connecting them with light sources or detector elements. A relatively new concept in this topic include self-written waveguides (SWWs), which have been intensively studied during the last decade [4–6]. Moreover, a connection between optical components, which are misaligned with respect to one another, is possible within limitations [7]. An advantage of such types of interconnection is the integrability into a variety of waveguide fabrication processes including hot embossing [8–13], photolithography [14,15], or direct laser ablation [16]. It offers the possibility to connect components that have been manufactured with different techniques, resulting in slightly different waveguide shapes [17]. Another technology that is able to create free form optics and thus can be used to connect different types of optical components is direct laser writing [18,19]. This technology is much more precise and flexible compared to the SWW-writing process, which on the other side

does not need expensive equipment or optical access to the position where the connection is to be established. Thus, SWWs are able to connect buried waveguides located inside a photonic integrated circuit, which cannot be easily achieved by other techniques [20,21].

An additional characteristic of adding an SWW in between waveguides or fibers that consist of other materials is the different optical response after changing external physical parameters, which is due to varying material characteristics. Concerning the thermo-optical coefficient (TOC), there are large differences between the optical materials. Thus, polymers show a large negative TOC, which can be 10–40 times higher than for conventional optical components such as glass [22]. This large value results in drift of the operating conditions when increasing or decreasing temperature. The change of the refractive index with respect to the temperature depends mostly on the change in density of the polymer due to the positive thermal expansion coefficient. Even small thermally-induced variations of the refractive index can significantly alter the intensity distribution within waveguides, especially in coupled systems [23]. Due to the material characteristics, polymers are thermally less robust than Si-based components. Depending on their respective glass transitions temperatures ($T_G$), polymers can be heated up to 60–250 °C (i.e., $T_G(PMMA) \approx 100\,°C$) and glass up to 400–700 °C (i.e., $T_G(BK7) \approx 570\,°C$), respectively [24–26].

In this work, we focus on the investigation of the thermal response of the SWW and whether this characteristic can be used as a sensing element. Therefore, detailed optical FEM-based and numerical simulation has been performed with different tools, and the results were compared with experimental measurements. We investigated theoretically and experimentally the variation of the refractive indices of an SWW by changing the thermal conditions and explain the underlying effects.

## 2. Sample Design

The focus of our work is the investigation of the thermal behaviour of an SWW which connects two silica fibers (OM2) with a core diameter of 50 µm. A scheme of the writing procedure of an SWW is shown in Figure 1.

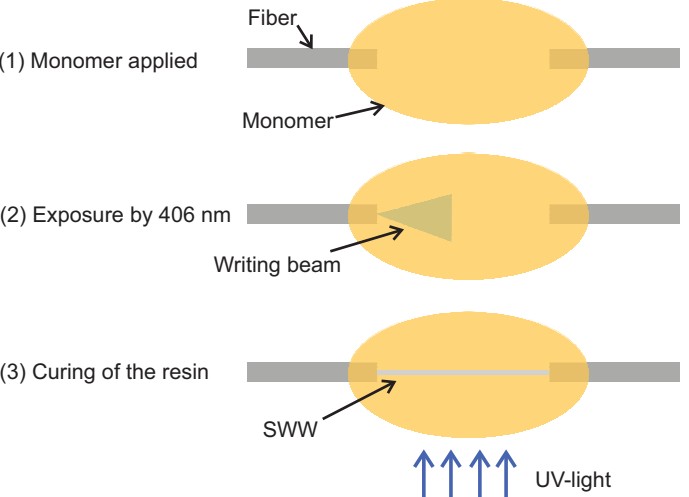

**Figure 1.** Scheme of the writing process of an SWW created between two silica fibers.

As indicated in Figure 1, a monomer drop is applied between two silica fibers which are aligned with respect to one another along their common optical axis. One of the fibers, which is connected to a laser diode, emits light at $\lambda = 406$ nm, which starts the polymerization process at the end of the fiber. This process step is illustrated in Figure 2.

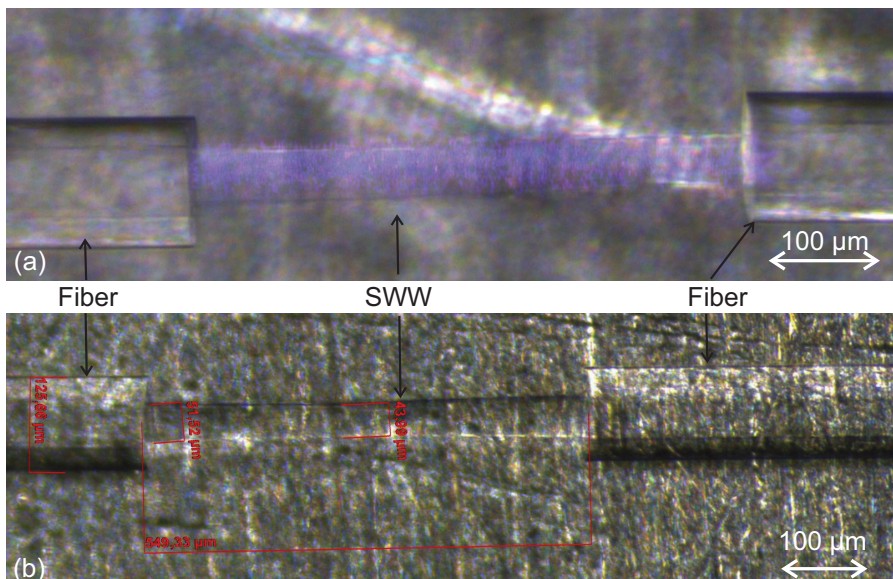

**Figure 2.** Image (**a**) is illustrating the creation of a SWW by a self-trapped laser beam with a wavelength of $\lambda = 405$ nm between two OM2-fibers with a core size of 50 µm and a cladding diameter of 125 µm, respectively. The red bars in the lower image (**b**) represents the measuring point for determining the thickness and length of the SWW. The fact that the width of the SWW decreases in the middle between the fibers is studied systematically as a function of the relevant process parameters in the next steps.

Due to the curing of the monomer, the refractive index increases locally at the point where the beam enters the monomer, which further traps the light and results in the development of a straight waveguide, as shown in Figures 1 and 2. For processing, it is advantageous to use multi-mode fibers. If a smaller fiber core size is chosen, it is more difficult to point the laser light to the opposite fiber end. Additionally, a smaller core size results in a smaller SWW diameter which subsequently reduces the adhesion strength and, thus, the robustness between the fibers and the SWW. In a last step the surrounding resin is cured by UV flood exposure. This enables a rigid connection between the two fibers. Here, it is important to realize that the final external curing yields two important characteristics. Due to the high material absorption of the UV light on the surface of the drop, the intensity for the curing inside the drop decreases significantly, which results in a refractive index distribution with a lower value inside the drop compared to its surface. This expected refractive index distribution is indicated in Figure 3.

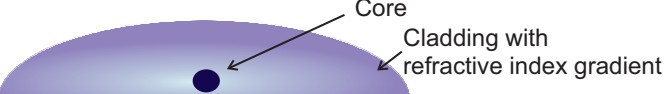

**Figure 3.** Cross section of a cured SWW with surrounding resin and gradually refractive index distribution. A darker color represents a higher refractive index.

The model of the refractive index distribution shown in Figure 3 is based on previous measurements where an approach based on Syntholux as matrix monomer was used [4]. The simulations and measurements presented in this work are based on NOA68, which is an optical adhesive from Norland Products. Due to the change of the material and the information needed for the simulation, the refractive index and the TOC have to be determined. The temperature-caused change of the refractive index is directly correlated to a change of the density and temperature itself. The whole correlation was determined as follows [27]:

$$\frac{dn}{dT} = \left(\frac{\delta n}{\delta \rho}\right)_T \left(\frac{\delta \rho}{\delta T}\right) + \left(\frac{\delta n}{\delta T}\right)_\rho = -\alpha \cdot \left(\frac{\rho \delta n}{\delta \rho}\right)_T + \left(\frac{\delta n}{\delta T}\right)_\rho \tag{1}$$

where $n$ is the refractive index, $\rho$ is the density, $\alpha$ is the volume coefficient of thermal expansion of a polymer, $dn/dT$ is the temperature-caused refractive index change, $(\delta n/\delta T)_\rho$ is the refractive index change for constant density, and $(\rho \delta n/\delta \rho)_T$ is a constant for a given polymer. Measuring all these parameters is very inconvenient and uncomfortable. The expected changes of the refractive indices are negative, which is common for polymers, and will be in the range of $\approx(-100--500) \times 10^{-6}$ K$^{-1}$, necessitating a very sensitive measuring device [28]. Here, the required refractive index measurements were performed by using a refractive index profilometer, which is based on the refracted near-field method, as schematically shown in Figure 4 [29].

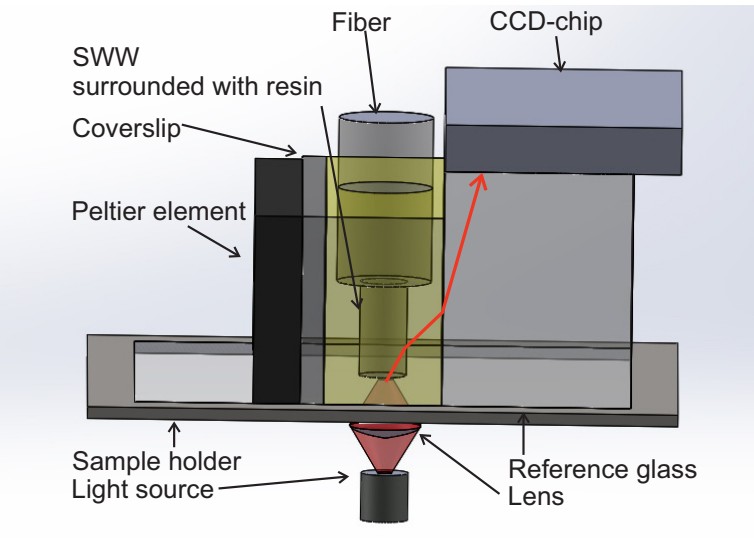

**Figure 4.** Scheme of the refractive index profilometer setup. A laser beam is focused on the sample surface. The main part of the beam exits the specimen through a layer of monomer (which is immersion oil in standard cases) on the backside of the reference glass, which is parallel to the entrance surface. Changes of the beam angle which exits the reference glass caused by a change of the refractive index in the beam path are detected by a large area CCD-Chip. The refractive index profilometer delivers a resolution of $\approx 10^{-5}$ and a pixel-resolution of $\approx 100$ nm at the employed wavelength of $\lambda = 638$ nm.

The near field approach allows the determination of the refractive index by measuring the change of the amplitude and deflection of the beam as given by the scalar wave function $n_{x,y}^2 \approx \frac{1}{k^2} \cdot \frac{\Lambda^2 A}{A}$, with $n_{x,y}$ as the two-dimensional refractive index profile, $k$ the wave vector, and $A$ as the field amplitude. The unknown refractive indices of the core and cladding will be approximated by using two references with defined refractive indices in the area of measurement, which is $\approx 500\,\mu m \times 500\,\mu m$. For this purpose, a reference glass block (i.e., BK7) is employed together with a specific immersion oil for which its refractive index is close to the expected one from the sample and the reference. Due to the measurement principle, it is not possible to determine the refractive index of the SWW if it is connected in between two fibers. Thus, for this measurement, a separate SWW was written perpendicular to the cover slit next to the reference glass as illustrated in Figure 4. In this configuration, the beam from the laser diode is focused on the upper surface of the cover slit, which is in contact with the sample. The focused beam is deflected by the different materials it propagates through, finally reaching the detector. Due to the determination of the TOC, we extended the measurement setup with a Peltier element to determine the refractive index of the sample at different temperatures. For this experiment, the immersion oil was exchanged by a thin coverslit which was characterized in detail beforehand. The results of measuring the refractive index distribution are shown in Figure 5.

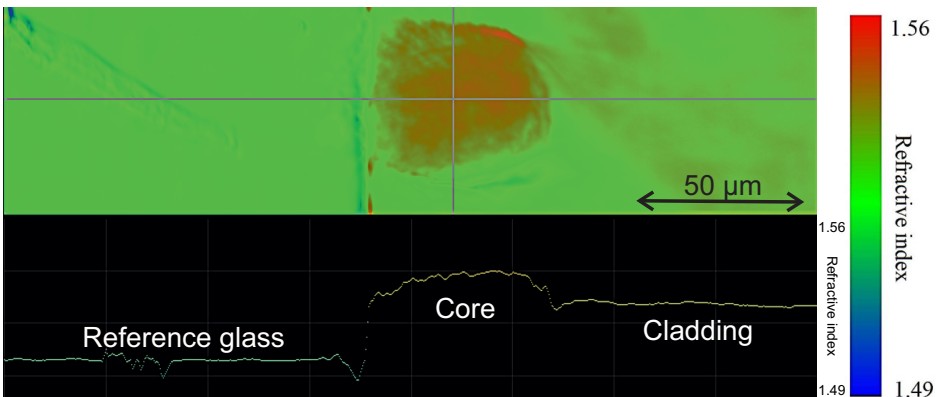

**Figure 5.** Measurement of the refractive index distribution of a SWW embedded in the cured cladding. The refractive index profile corresponds to the cross section along the horizontal line in the false colour image. The y-axis provides the refractive index value.

The depiction in Figure 5 shows the higher refractive index of the SWW core and its gradient distribution. Due to the integration of a Peltier element as shown in Figure 4, the refractive index profilometer is enabled to measure the refractive index distribution at different temperatures in the range of 25–65 °C. The measurements taken with this device yields the required values for the refractive indices of the SWW core and cladding, which are required for the simulation. The somewhat irregular shape of the SWW shown in Figure 5 is due to the measurement setup. The depicted end facet is directly attached to the cover slit (see Figure 4), which results in reflections from the glass during the writing process, resulting in unwanted curing at the borders. A slight tilting can also not be excluded, which affects the measurement accuracy.

## 3. Theoretical Investigation of Thermal Behaviour of SWWs

The goal of this investigation is a better understanding of the interconnecting element, especially concerning the thermal response. We expect a separable change of the optical transmission of the fiber-SSW-fiber assembly in comparison to two fibers only due to its multi-material structure consisting of a transition from glass (fiber) to polymer with different material properties caused by the varying curing procedures.

The required material parameters for the simulation are the TOC and refractive indices of the core and cladding. These measurements have been conducted for NOA68, with the setup depicted in Figure 4. The obtained results are shown in Table 1.

**Table 1.** Average values of the refractive index measurements (at $\lambda = 638$ nm) at different temperatures and their calculated $\Delta$ and g-parameters required to calculate the refractive index of the SWW.

| T [°C] | $n_{core}$ | $n_{cladding}$ | $\Delta$ | g |
|--------|------------|----------------|----------|-----|
| 25 | 1.51845 | 1.51767 | 0.000128 | 1.99969 |
| 30 | 1.51826 | 1.51710 | 0.000190 | 1.99954 |
| 35 | 1.51819 | 1.51690 | 0.000212 | 1.99949 |
| 40 | 1.51808 | 1.51674 | 0.000221 | 1.99947 |
| 45 | 1.518 | 1.5165 | 0.000240 | 1.99942 |
| 50 | 1.51786 | 1.51632 | 0.000253 | 1.99939 |
| 55 | 1.51777 | 1.51611 | 0.000272 | 1.99935 |
| 60 | 1.51767 | 1.51590 | 0.000291 | 1.99930 |

The measured refractive indices show a decreasing tendency with increasing temperature for core and cladding. The additionally shown $\Delta$ values are defined by $\frac{n_{core}^2 - n_{cladding}^2}{n_{core}^2}$. The last column of Table 1 contains the calculated g-parameter. Both, g-values and $\Delta$-values allow the calculation of the refractive index distribution of the SWW based on the following:

$$n(r) = n_{core}\sqrt{1 - 2\Delta(\frac{r}{a})^g} \tag{2}$$

for $r \leq a$ where $a$ is the size of the core. For the sake of completeness, the refractive index distribution can be calculated as $n_r = n_{core}\sqrt{1 - 2\Delta}$ for $r > a$, which results in a step-index distribution for $g \rightarrow \infty$. G-values of $\approx 2$ for all measurements result in a gradual change of the refractive index from the center towards the cladding. The calculated refractive index distribution and change with the temperature are shown in Figure 6.

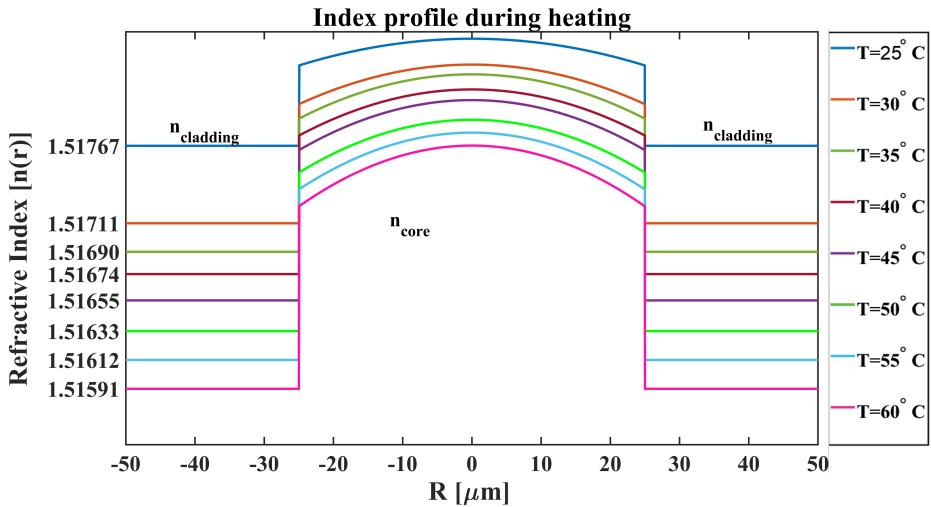

**Figure 6.** Refractive index distribution of core and cladding of the SWW during the heating process showing a decreasing refractive index with increasing temperature. Moreover, the refractive index difference between core and cladding increases at higher temperatures.

The expected refractive index distribution shown in Figure 6 was confirmed by measurements taken with the refractive index profilometer, being obvious in the shape of the core of the cross-sectional view of Figure 5.

Further investigations aimed at the quantification of the change of the optical transmission during the heating of the SWW. The simulation was fed with the measured refractive index values for each temperature and the corresponding TOC. The obtained results here were in contrast to our expectations, where the transmitted optical intensity should decrease with the temperature and which was expected due to previous simulations based on the literature values and a constant TOC for core and cladding [30]. The results obtained in the simulations for the new case are shown in Figure 7.

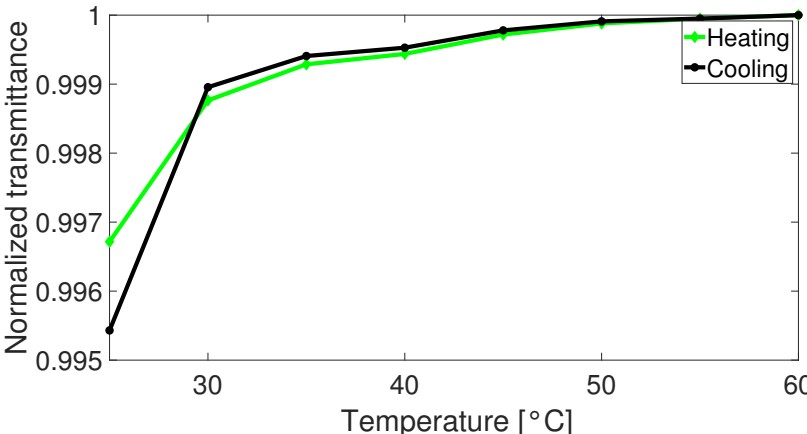

**Figure 7.** Simulation of the change of the optical transmission with increasing temperature for the heating and cooling process applied to the SWW.

An explanation of this behaviour lies in the different TOCs. These values are normally defined as $dn/dT$ describing the change of the refractive index of a material with the temperature. This parameter, again, is defined (in linear approximation) as $\frac{dn}{dT} = [\frac{\lambda}{2L\Delta T}] - n\alpha$ [31], with $\lambda$ as corresponding wavelength, $L$ as sample thickness, and $\alpha$ as linear expansion coefficient. In particular, the latter is a material specific parameter which depends on the density, among others. As obtained from previous experiments, the refractive index and the density, respectively, correlate with the curing time and intensity [4]. Additionally, the fact that the monomer and cured polymer show a high absorption coefficient results in a refractive index gradient of the surrounding resin, which was cured with UV flood exposure. This enables that the surface of the resin is completely hardened, whereas the core-near region is also solid but less hardened, depending on the curing time and thickness of the resin applied. This, in sum, enables us to explain the achieved simulated results. The different material characteristics due to the varying types of curing result in an increasing refractive index difference between core and cladding as discernible from the values of Table 1. Subsequently, this fact results in an increase in the NA of the waveguide when heating is applied, as indicated in Table 2, based on the calculation NA = $\sqrt{n_{core}^2 - n_{cladding}^2}$.

**Table 2.** Change of the numerical aperture with increasing temperature calculated from the refractive indices of Table 1.

| T [°C] | NA |
|--------|------|
| 25 | 0.049 |
| 30 | 0.059 |
| 35 | 0.063 |
| 40 | 0.064 |
| 45 | 0.067 |
| 50 | 0.068 |
| 55 | 0.071 |
| 60 | 0.073 |

The obtained values shows an increase in the NA $\approx 67\%$ during the heating cycle up to 60 °C. This decreases the coupling losses from the silica fiber to the polymer structure. In comparison, the used optical fibers have a numerical aperture of NA $= 0.2 \pm 0.015$ (OM2-fiber), which accepts a higher number of modes than the SWW.

In order to start experimental validation with the setup shown in Figure 8, we studied the temperature distribution within the material numerically in order to verify that the temperature obtained correlates with the one of the SWW. Therefore, we simulated the

distribution of the temperature by heating an SWW surrounded by the cured cladding using the COMSOL Multiphysics package. The result of this investigation is shown in Figure 9.

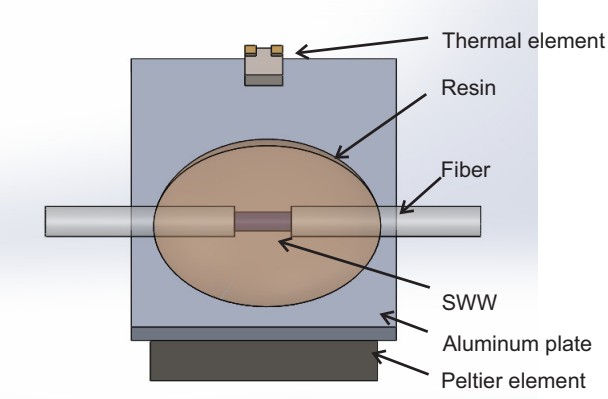

**Figure 8.** Experimental setup for verifying the simulated thermal behaviour of the SWWs. The interconnecting structure is written in between two multi-mode silica fibers (OM2) on top of an aluminum plate, which is heated by a Peltier element (QC-32-0.6-1.2, Quick-Cool). The actual temperature is checked with an additional thermal sensor (Heraeus Nexensos M222).

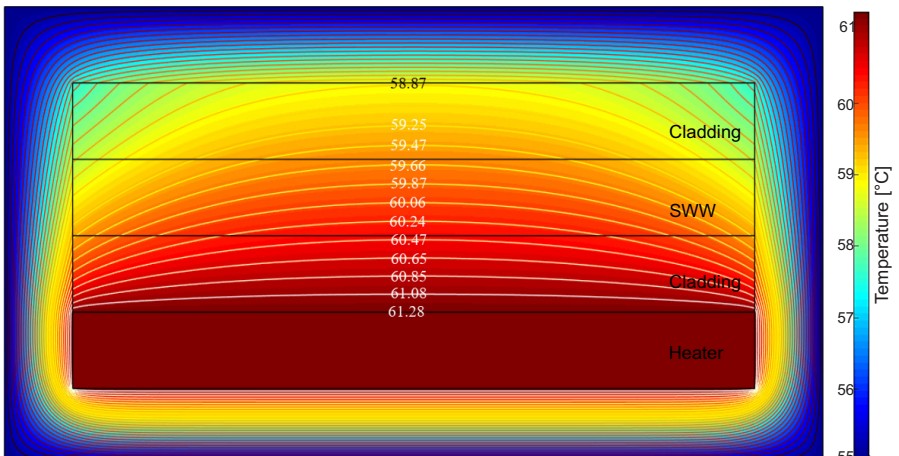

**Figure 9.** Simulated contour map of the temperature distribution of the cross-section in the middle of an SWW when the heater was at maximum temperature of 61.28 °C. The thickness of each layer is assumed as 50 µm, which corresponds to the thickness of the fiber core we normally use for the SWW experiments. Due to alignment purposes, the SWW should be very close to the bottom so that the thickness of the lower cladding with 50 µm is a realistic assumption. The mesh size used for this simulation was ≈3 µm.

For the simulation shown in Figure 9, the heater bottom layer (heater) applies the thermal energy, which distributes to different layers and the surrounding air is working as a heat sink. In the case of cooling, the heat input from the bottom layer was set to zero, and the material cooled down by heat exchange with the surrounding air. The simulation yields the temperature distribution inside the SWW for the case of heating the structure up to 60 °C. By using this simulation, we also examined the time it takes until the SWW reached the expected temperature. These results are listed in Table 3.

**Table 3.** Values to be set for the heater to reach the desired temperature of the SWW in the simulation.

| Heating | | |
| --- | --- | --- |
| $T_{Heater}$ [°C] | $T_{SWW}$ [°C] | $t$ [s] |
| 26.46 | 25 | 1.15 |
| 31.42 | 30 | 0.83 |
| 36.43 | 35 | 0.83 |
| 41.4 | 40 | 0.81 |
| 46.39 | 45 | 0.81 |
| 51.37 | 50 | 0.93 |
| 56.32 | 55 | 0.79 |
| 61.28 | 60 | 0.8 |
| Cooling | | |
| $T_{Heater}$ [°C] | $T_{SWW}$ [°C] | $t$ [s] |
| 61.28 | 60 | 0 (start) |
| 56.32 | 55 | 1.76 |
| 51.37 | 50 | 1.77 |
| 46.39 | 45 | 1.78 |
| 41.4 | 40 | 1.78 |
| 36.43 | 35 | 1.78 |
| 31.42 | 30 | 1.79 |
| 26.46 | 25 | 1.79 |

Due to the small thickness of the sample, the entire material achieves the desired temperature nearly instantly after setting the values for the heater.

## 4. Experimental Verification

In order to verify the unexpected behaviour of the SWWs as function of temperature, i.e., the increase (instead of decrease) in the transmitted intensity with increasing temperature, an experimental setup was used based on a Peltier element (QC-32-0.6-1.2, Quick-Cool), and it was used to heat the sample. An applied thermal sensor (Heraeus Nexensos M222) was employed to control the temperature. The SWW was created at room temperature in between two silica fibers on top of an aluminum plate, which was attached to the Peltier element. The writing and adjustment process is performed manually which may result in small differences in the performance for different experiments. The normal deviations between different performances are very small (negligible) if a trained user is running the experiment. An automation of this process based on image analysis and auto-alignment may also be possible and is likely to improve the repeatability. As UV-curable adhesive NOA68 (Norland Adhesive) was used. One of the fibers is connected to a multi-channel laser diode module (MCLS1, Thorlabs), which emits laser radiation at $\lambda = 406$ nm for the writing procedure and $\lambda = 638$ nm and $\lambda = 850$ nm for characterizing the structures. The required external curing was performed by using a germicidal lamp with an intensity of $\approx 1.8$ mW cm$^{-2}$ at 254 nm (40-VL-208 G, Bio-Budget). The experimental setup is shown in Figure 8.

The difference between the simulated setup and the experimental configuration shown in Figure 8 is the aluminum plate, which protects the heater surface against the monomer. This plate is not considered in the simulation. However, we experimentally measured the temperature on top of the aluminum plate, which is our reference layer for setting the temperature of the waveguide core.

The measurement of the change of the optical transmission with the setup sketched in Figure 8 was achieved by using two different fiber-coupled light sources emitting laser radiation at $\lambda = 638$ nm and $\lambda = 850$ nm, respectively, and a fiber-coupled photodetector (S151C, Thorlabs) on the detection side. The used Peltier element enables us to measure

the optical transmission within a temperature range of 25–60 °C. With this setup, it was also possible to study both expected behaviours of the SWW during the heating, which are the cases of decreasing and increasing optical transmissions, respectively, depending on the curing time applied during the waveguide fabrication. The achieved results of these measurements are sketched in Figure 10.

As mentioned already, the reason for the opposite behaviour of the SWWs shown in Figure 10 is that different curing times of the external UV flood exposure were applied. Due to slightly different drop sizes of the resin, it was not possible to define a sharp threshold where the thermal response of the SWW switches from a decreasing (Figure 10 top) to an increasing (Figure 10 bottom) optical transmission as a function of temperature. An external curing time of more than 20 min usually results in complete curing of the droplet. In the other case, a curing time of less than 10 min, usually just a few minutes (1–2 min), is sufficient for solidifying the surface only and created a rigid connection as well as a refractive index gradient inside the material. These experiments were all performed with the same UV-lamp for the external curing and may differ for light sources with a higher fluence.

The experimental results show that we are able to adjust the physical parameters of the polymer waveguide core and cladding and the corresponding optical behaviour. A more detailed investigation of this effect with different lengths of the SWWs resulted in different temperature sensitivities, indicated by the different slopes in Figure 11.

The measurements for the different gap sizes shown in Figure 11 reveal that the gap size also affects the thermal response of the SWWs where an increasing gap size results in an increasing change of the optical transmittance.

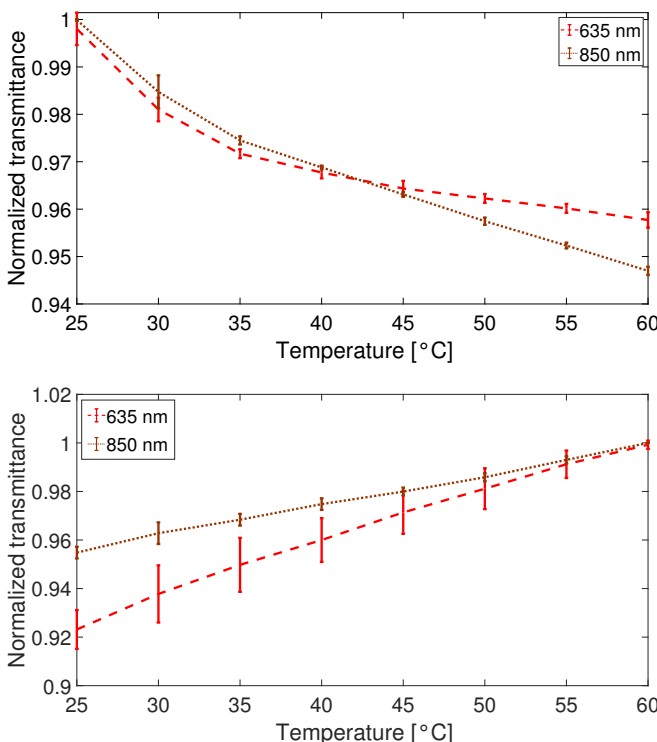

**Figure 10.** Experimental results from the thermal response of the SWW heated from room temperature to 60 °C. Depending on the external curing time, we were able to measure a decreasing (**top**) or increasing (**bottom**) optical transmission as the temperature increased.

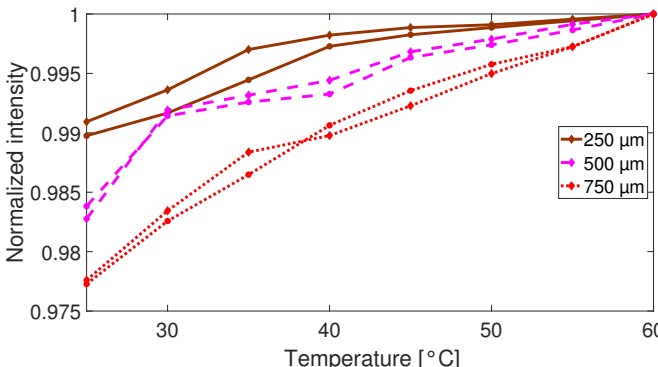

**Figure 11.** Thermal response of the SWWs for different gap sizes of 250 μm (brown line), 500 μm (dashed magenta line), and 750 μm (dotted red line). The measurements were performed with the diode laser at $\lambda = 850$ nm. The two lines for each color corresponds to the heating and cooling cycle, respectively.

## 5. Conclusions

In this work, we investigated the unusual behaviour of self-written waveguides as a function of temperature and different curing times applied during the waveguide production. The observations support the assumption that the behaviour is due to a change of the material parameters during the external curing phase. Therefore, the TOCs for the core and cladding became different, which finally increases the numerical aperture NA $= \sqrt{n_{core}^2 - n_{cladding}^2}$ during the heating process. We also noticed a difference in the change of the transmitted intensity during the heating process between the simulation and the experimental results, see Figures 7 and 11 (for the gap size of 500 μm ). This is due to the missing material parameters which are required for the simulation but are not available, i.e., the complex refractive index parameter $k$, mechanical parameters as the linear expansion coefficient, or the material density for different curing times. Therefore approximations were necessary, or values from similar materials were taken, respectively. The achieved temperature sensitivity of $\approx 0.4\%\ \mathrm{K}^{-1}$ is relatively small. However, the fact that the change of the signal is opposite to the one induced by other environmental influences such as, i.e., drift of the intensity of the light source, degradation of the waveguides, or change of the coupling conditions is an interesting property. Generally, the structure is integratable in more complex photonic networks and could be used as sensing element therein.

Possible further steps for increasing the performance of this structure to be employed as a temperature sensor are, e.g., the use of materials with more favorable thermal behaviour or the exchange of the cladding material by removing the liquid resin after the SWW writing. Therefore, more detailed simulations are required to evaluate which material combinations are usable to increase the temperature sensitivity of the structure. Another point of interest is the optimal combination of the waveguides core size. Due to the fact that increasing temperature changes the NA and, thus, the optical transmission of the structure, a combination of different core diameters for input and output fibers can achieve a change in the optical transmission in the same manner, which can even be larger than the observed effect so far.

The unique thermal response of the structure opens a promising new field of applications. In this work, we characterized a single external parameter in detail, which affects the performance of the SWW. Future work will focus on the investigation of the influence of humidity and axial strain on the SWW transmission. These studies might enable the reliable detection of multiple parameters simultaneously.

**Author Contributions:** Conceptualization, A.G.; methodology, A.G. and M.B.; investigation, M.B. and A.G.; resources, B.R.; data curation, A.G.; writing—original draft preparation, A.G.; writing— review and editing, A.G., M.B., B.R. and W.K.; supervision, B.R. and W.K.; project administration, A.G. and B.R. All authors have read and agreed to the published version of the manuscript.

**Funding:** This research was funded by Deutsche Forschungsgemeinschaft (DFG, German Research Foundation) under Germany's Excellence Strategy within the Cluster of Excellence PhoenixD (EXC 2122, Project ID 390833453).

**Institutional Review Board Statement:** Not applicable.

**Informed Consent Statement:** Not applicable.

**Data Availability Statement:** Data available on request due to restriction of privacy. The data presented in this study are available on request from the corresponding author. The data are not publicly available due to related experiments are still in progress and involve unpublished papers.

**Conflicts of Interest:** The authors declare no conflict of interest.

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
