# Peer review of "Simulation and Experimental Verification of the Thermal Behaviour of Self-Written Waveguides"

_applsci, doi:10.3390/app11177881_

Round 1

Reviewer 1 Report

In this submission, a very facile and low-cost method is developed to realize quick construction of waveguide devices as the connector between two optical fiber ends. The refractive index and its profile was tested quantitatively regarding the spatial distribution and temperature dependance. The device was proved to work as a temperature sensor. In the current state, the experiments seem to be not quite sufficient for a project on an optical device, so does the novelty. Before being considered for publishing, I have some comments or questions as followed.

  1. The macro and micro characterizations of as formed devices should be added such as optical and electronic microscopies to directly show their morphology and quality.
  2. How is the reproducibility of this method of waveguide fab among devices? Compared with micro/nano-fabrications like laser direct write [e.g., Small 11, 2869 (2015)] and lithographies? A better introduction on the novelty and a comparison with other fabrication techniques for optics should be of great help, as well as a comparison with other temperature optical sensors [e.g., IEEE Sensor J.15, 3429 (2015).].
  3. Do the diameter and the distance of the fibers influence the fabrication significantly? How?
  4. The output light from a fiber end diffuses in a space before the next fiber. Should this unfocused and non-parallel light beam form a non-parallel structure of polymerized photoresist? Is the cross-section diameter of the fabricated waveguide constant along the optical axis?
  5. What is the size of the waveguide fabricated? A scale bar should be added in Fig. 4.
  6. Based on Fig 4 of the refractive index profile, the cross section of the fabricated waveguide is irregular.
  7. Please describe the test of refractive index with more details. The RI profiling of the fabricated device connected with two fibers is interesting. Figure 3 can be improved together with Figure 8 for a vivid schematic.

Author Response

Dear Reviewer,

thank you very much for your comments. Please see the attachments for the corrected (red lined) version and our response to the reviewers comments.

Sincerely,

Axel Günther

Reviewer 2 Report

The article presents both simulation and experiment results of a sww and claims that the curing time could change the transmission properties w.r.t. temperature. The finding is interesting and has potential impact to the community. However, I found the article is quite hard to follow due to lack of information and poor presentation. Therefore, substantial rewriting of the article is necessary before it is acceptable for publication. Here are a few comments:

  1. Eq. (1) needs to be reformatted, the subscripts "T" and "rho" are indistinguishable to normal symbol "alpha".
  2. Fig. 1, what type of fiber is used? What's its core/cladding diameter and refractive indices? Is it single mode or multimode fiber?
  3. What's the operating temperature during curing? Does that correlate the TOC discussions in the rest of the article?
  4. I am having hard time to correlate Fig. 3 with Fig. 2 & 4. Is the monomer showing in Fig. 3 just bulk material or already fabricated to sww through the procedure in Fig. 2? If is the latter, show the orientation of the fiber in Fig. 3 and the thickness of the monomer.  Also show the scanning range of Fig. 3 that corresponds to Fig. 4. A 3D plot of Fig. 3 may help a lot if the authors can draw it. 
  5. Add x and y axis on Fig. 4 (actually should be 2 y axis, one on top plot showing physical height of the measurement, one on bottom plot showing the refractive index values) so that the readers know the size of the scanning area.
  6. What's the pixel resolution, laser wavelength and measurement sensitivity of the profilometer?  
  7. How Eq. 2 agrees with the measurement results? The authors should plot Fig. 5 along with the measured results (maybe reduce the number of temperature points to 4 so that the plot is not too busy).  
  8. The author should also list the RMSE between the Eq. 2 fitting and the measurement results in Table 1.
  9. Fig. 6 looks like the heating and cooling results are quite consistent except at around 25C, could the authors comments why the results at that point shows large difference? 
  10. The caption in Fig. 6 says the simulation is doe "at different wavelengths", couldn't see anything related to wavelength in the plot and main text.
  11. In fact, why the simulation of Fig. 6 will show that transmission difference between heating and cooling at all?
  12. Fig. 7, what's the boundary condition used in the simulation? Is that realistic? What's the physical length/width of the computation window? What's the thickness of each layer? Wouldn't SWW core be circular in shape according to Fig. 4? Given the fact that the heater is much larger than the size of drop, I can hardly believe there is a temperature gradient along horizontal direction. It appears to be spurious due to a wrong boundary condition.
  13. Line 107, add a citation to the formula.
  14. What's the wavelength used in Table 2, again is it correlated to the curing temperature? Also what's the NA of the fiber used? Without it larger NA of the SWW not necessarily means larger transmission.
  15. Table 3, again how reliable are these simulation results? The authors should verify them with the experiments (ie. measure the core refractive index vs time use their setup in Fig. 3.)
  16. Fig. 9 add simulation results to these two plots and compare.
  17. Again no mention on the fiber type in the experiment part.
  18. Fig. 10 caption, I see "250 m", "500 m", "750 m"...
  19. Fig. 6, 9, 10 change the y label to "Transmittance" would be clearer.

Overall the topic of this article is good but the authors should spend much more time to carefully write this article.

Author Response

Dear Reviewer,

thank you very much for your comments. Please see the attachment for the response to your comments and the updated (red lined) version.

Sincerely,

Axel Günther

Round 2

Reviewer 1 Report

I think the authors have answered the comments or questions very well. It could be published in the current form.

Reviewer 2 Report

The authors have addressed all comments.